# Text to Blind Motion

**Hee Jae Kim**[1]  **Kathakoli Sengupta**[1]  **Masaki Kuribayashi**[2]

**Hernisa Kacorri**[3]  **Eshed Ohn-Bar**[1]

[1]Boston University  [2]Waseda University  [3]University of Maryland, College Park

## Abstract

People who are blind perceive the world differently than those who are sighted, which can result in distinct motion characteristics. For instance, when crossing at an intersection, blind individuals may have different patterns of movement, such as veering more from a straight path or using touch-based exploration around curbs and obstacles. These behaviors may appear less predictable to motion models embedded in technologies such as autonomous vehicles. Yet, the ability of 3D motion models to capture such behavior has not been previously studied, as existing datasets for 3D human motion currently lack diversity and are biased toward people who are sighted. In this work, we introduce BlindWays, the first multimodal motion benchmark for pedestrians who are blind. We collect 3D motion data using wearable sensors with 11 blind participants navigating eight different routes in a real-world urban setting. Additionally, we provide rich textual descriptions that capture the distinctive movement characteristics of blind pedestrians and their interactions with both the navigation aid (*e.g.*, a white cane or a guide dog) and the environment. We benchmark state-of-the-art 3D human prediction models, finding poor performance with off-the-shelf and pre-training-based methods for our novel task. To contribute toward safer and more reliable systems that can seamlessly reason over diverse human movements in their environments, our text-and-motion benchmark is available at https://blindways.github.io/.

## 1 Introduction

Computational modeling of people and their 3D motion has been studied extensively by the machine learning community over the past decades [2, 11, 12, 25, 44, 49, 54]. More recently, the field has been moving beyond single actors performing contrived actions to model interactive behaviors, *i.e.*, incorporating objects and surrounding people. However, the scope and diversity of the datasets associated with this prior work remain limited to a simulation [53], a lab [27, 43], or simplified layouts [19, 23, 28, 40].

Efforts for accurately capturing and modeling natural and subtle 3D human motion in more realistic setups, such as [1, 57, 65, 66], are still lacking in complex and safety-critical urban scenes that comprise dynamic intersections, intricate layouts, and dense social settings. Even more noticeably, there has not been a single 3D human motion dataset released that comprises mobility data from individuals with disabilities. Hence, while most human motion models are developed with assistive and interactive applications in mind, such as social robots and autonomous driving, those who could benefit the most from these technologies are not included. Such severe biases in existing benchmarks can carry broader societal implications. It can exacerbate already widespread concerns in accessibility, where autonomous vehicles fail to accurately predict and safely respond to movements of people with disabilities [64]; a population that is already disproportionately impacted by motorists' lack of awareness [21, 36, 37, 46]. In this work, we address this current gap in literature through a novel benchmark featuring people who are blind navigating real-world urban settings.

38th Conference on Neural Information Processing Systems (NeurIPS 2024) Track on Datasets and Benchmarks.

Blind pedestrians are known to exhibit significantly different mobility characteristics based on personal factors, such as their lived experiences with disability or the use of mobility aids (*e.g.*, canes, guide-dogs, and orientation and mobility apps [29]). For instance, many do not face forward to signal intent to cross before stepping into the road and may take longer to explore tactile cues when crossing in various intersections [4, 13, 20, 18, 21]. Additionally, some may veer significantly in open spaces or unexpectedly step into the road due to obstacles, such as a truck parked at obstructed intersections with damaged or ambiguous curbs. In such scenarios, reasoning over subtle 3D behaviors, like hand-aid coordination gestures, could improve future prediction in autonomous vehicles and avoid potential safety-critical outcomes [62].

Yet, as far as we are aware, no prior work has investigated the prediction of pedestrian motion in such edge cases, characterized by inherently distinct, subtle, and uncertain nature. Specifically, we aim to understand the capabilities of state-of-the-art 3D motion models for modeling and predicting future motion of blind pedestrians—ultimately, to ensure that autonomous systems and vehicles in urban environments operate safely around pedestrians with disabilities.

**Contribution:** Our overarching goal is to enable more robust, accurate, and needs-aware pedestrian behavior prediction models that effectively account for disability-related scenarios and behaviors. Our key contribution is twofold. First, we introduce BlindWays, a novel multi-modal 3D human motion dataset featuring pedestrians who are blind navigating unfamiliar and complex real-world environments. Our dataset includes detailed language-based annotations of context and non-visual navigation strategies. Second, we use the dataset to benchmark state-of-the-art 3D motion models within our novel modeling task. By analyzing the effects of model pre-training and fine-tuning on future motion prediction, we identify fundamental limitations in the generalization of current datasets and models, particularly when evaluated within diverse and rare human attributes.

## 2   Related Work

To design our study and data collection, we build on recent advances in in-the-wild 3D human motion estimation, 3D human motion models, and language-based motion generation.

**Estimating In-the-Wild 3D Human Motion:** Researchers have long sought to capture 3D human motion in naturalistic settings [7, 38, 39, 42, 50, 55, 56, 60, 65, 68]. However, vision-based inference of 3D keypoints can be unreliable in our context of multi-actor, dense scenes with frequent occlusion and interaction with objects [71]. For instance, AlphaPose [10], a widely used keypoint detection and lifting model, exhibited frequent failure and poor performance on videos for our settings (online sourced and our in-house collected ones [33]). Given the difficulty in manually annotating monocular videos with accurate 3D information [40], an alternative approach for minimally intrusive collection involves wearable inertial sensors [9, 65, 69, 79, 48]. Specifically, in our study, we leverage an Xsens [69] set of trackers (one placed on the mobility aid). While the system may have inherent noise, it can be re-calibrated to improve accuracy. In our study, we frequently re-calibrate the system throughout the route to minimize drift and improve accuracy. We also filter out noisy skeletons through manual inspection in scenarios of system tracking failure. Nonetheless, we emphasize that motion capture in-the-wild remains an open challenge.

**Modeling 3D Human Motion:** Motion-prediction models (*e.g.*, [33, 41, 54, 70, 72, 73]) do not currently model pedestrians with disabilities [30, 47, 75]. The few studies that have analyzed blind navigation motion in context are qualitative in nature [4, 18, 20, 21], only providing a high-level account. Instead, 3D motion models generally leverage sighted participants [6, 15, 24, 26, 31, 65, 67, 78]. While understanding and encompassing unique navigation behaviors is essential for a comprehensive motion synthesis and generation frameworks [15, 16, 81], our study empirically demonstrates how prior works struggle to generalize to the nuanced modeling of blind motion.

**Text-to-Motion:** Text-driven motion generation has gained significant interest due to its controllability, as well as the concise context information provided by textual descriptions. Diffusion models, such as MDM [61], have been explored for generating human motion sequences from text descriptions, progressively refining the motion through a series of forward steps. However, recent diffusion-based approaches [5, 28, 61, 76] do not generate plausible blind motion, as shown by our study. In addition,

Table 1: **Comparing Motion Benchmarks.** BlindWays introduces several dataset dimensions not explored by prior work, including participants with mobility aids (*i.e.*, white cane or guide-dog, tracked with a sensor) and safety-oriented scenarios in urban streets. We also provide language annotations with two levels of granularity: high-level summaries and more detailed low-level descriptions.

| Dataset | Participants | | | Motion Data | | Text Annotation | | Context | | |
| --- | --- | --- | --- | --- | --- | --- | --- | --- | --- | --- |
| | Number | Disability | Aid | Hours | Source | Mean Length | Two-Level | Outdoor | First-Person Cam. | Safety |
| Human3.6M [27] | 11 | ✗ | ✗ | 2.9 | Marker-based | - | ✗ | ✗ | ✗ | ✗ |
| AMASS [43] | 344 | ✗ | ✗ | 40.0 | Marker-based | 11.9 | ✗ | ✗ | ✗ | ✗ |
| HumanML3D [15] | 450 | ✗ | ✗ | 28.6 | Marker-based | 12.3 | ✗ | ✗ | ✗ | ✗ |
| Motion-X [40] | - | ✗ | ✗ | 144.2 | Marker-based | 38.5 | ✗ | ✓ | ✗ | ✗ |
| **BlindWays (Ours)** | 11 | ✓ | ✓ | 2.8 | IMU-based | 44.1 | ✓ | ✓ | ✓ | ✓ |

current methods are limited by inefficiency during testing, as they require multiple forward steps to generate a single motion sequence. GPT-based text-to-motion models [15, 16, 81] have recently shown promising results. TM2T [16] focuses on the temporal modeling of motion, ensuring that generated motions are coherent and contextually appropriate. Recently, MotionGPT [28] has integrated generative pre-trained transformers with joint training of motion and language, further advancing the quality and diversity of generated motions. However, the applicability and generalization of such models in accessibility settings remain underexplored.

**Motion and Language Benchmarks:** Datasets with high-quality text descriptions have further advanced the controllability and generation of multimodal motion, with some also incorporating interactions with objects and other people. While motion-language models have recently achieved outstanding performance in tasks such as motion prediction [3, 74], diverse motion generation [15, 17, 43, 51, 52], and the study of human-object interaction [22, 23, 58, 77, 80], researchers have been inspired to develop diverse datasets that support these varied motion tasks. The KIT-ML [51] dataset focuses on multi-modal language-to-motion translation but lacks motion diversity. AMASS [43] unifies a wide range of MoCap data. BABEL [52] and PoseScript [8] also incorporate action labels and textual descriptions, however, such datasets still lack in motion diversity and realism. HumanML3D [15] introduces a large collection of 3D human motions with corresponding natural language annotations, primarily focused on static indoor settings with repetitive motions. Motion-X [40] introduces a comprehensive dataset that includes detailed semantic annotations and outdoor environments (a context largely neglected in previous datasets). However, the distinct and uncertain nature of blind pedestrians' motion remains unexplored, limiting the generalizability of state-of-the-art motion modeling models in these critical cases, despite their importance. Our work aims to address this gap by introducing BlindWays, providing a richer and more challenging dataset for generalizing text-to-motion models and capturing the subtle movements of the head, limbs, and mobility aids used by blind individuals. Similar to Motion-X, BlindWays dataset is collected entirely in outdoor settings and incorporates IMU-based motion capture, unlike traditional marker-based systems [32, 35] and is less restricted by environmental constraints, allowing motion tracking in diverse outdoor settings.

## 3 The BlindWays Dataset

### 3.1 Overview

We collected BlindWays, a comprehensive blind motion dataset comprising 1,029 motion clips and approximately 0.6 million human poses, along with 2,058 detailed, paired, high- and low-level text descriptions. We capture natural motion data from 11 blind and low-vision individuals navigating dynamic outdoor environments along carefully engineered paths exhibiting various challenges. Notably, this is the first work to propose blind motion data enriched with text descriptions, an exceptionally challenging and labor-intensive process. BlindWays's text descriptions are informed by third-person and egocentric videos, each totaling 0.3 million frames. Specifically, captured contextual videos play a critical role in the annotation process by providing an overall scene of blind motion, allowing annotators to sufficiently leverage scene and video context to accurately, precisely, and expressively describe the motion. To synchronize between motion data and videos, we asked participants to clap at the beginning of each route. To ensure high quality, the MoCap system is calibrated in each route and text descriptions are annotated in-house by human annotators, including motion experts, and are carefully checked. We employ a wearable IMU-based system and filter noisy sequences to maintain accuracy and reliability.

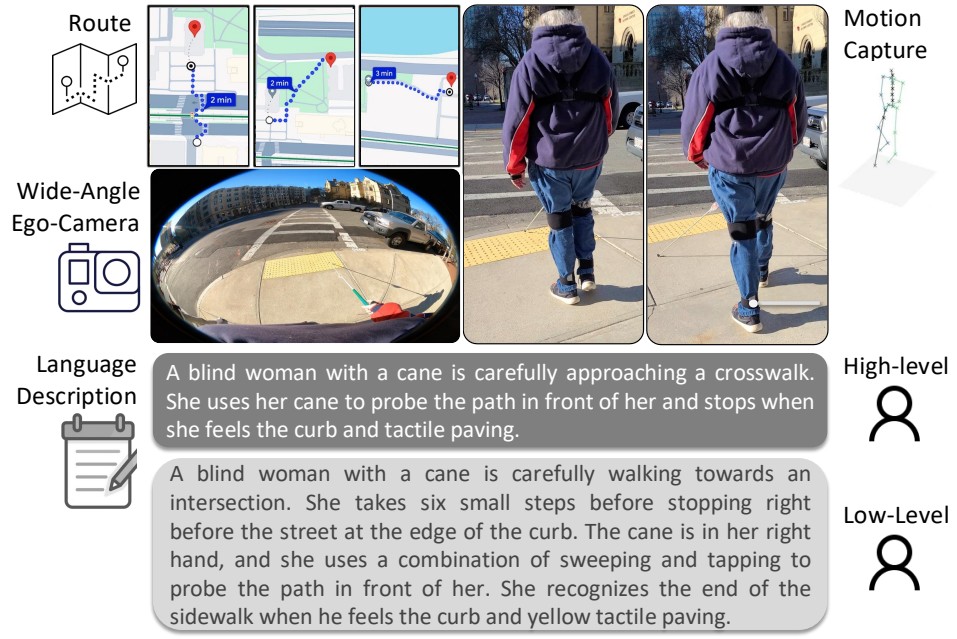

Figure 1: **Data Collection with Wearable IMU-based Sensors.** Depicting a frame from the study with diverse route stimuli, intersections, a motion capture, and a wide-angle egocentric camera view.

## 3.2 Data Collection Procedure

We conducted a user study involving 11 participants, consisting of three women and eight men, all of whom are either blind (N=10) or have low vision (N=1). Each participant utilized their own mobility aid, which included either a cane or a guide dog, to record natural behavior. Our participants represent a diverse range of ages, levels of visual impairment, and mobility aids, ensuring a rich data collection of navigation behaviors (details on participant demographics are provided in our supplementary). Participants are equipped with bone conduction headsets to receive real-time auditory instructions from Google Maps. Our data collection was approved by our Institutional Review Board. Each participant provided informed consent before participating in the study and was compensated $50/hour for up to three hours, including travel time; data collection sessions typically lasted two hours or less. We note that two researchers always followed the participants during dataset collection to ensure their safety.

**Scenarios:** In collaboration with local blind advisors and sighted certified orientation and mobility instructors, we engineered eight distinct routes to encompass a variety of real-world scenarios that blind people commonly encounter. These scenarios include walking on the curb, crossing streets, navigating open spaces, and ascending and descending staircases. For example, while crossing streets, participants faced a challenge when encountering a subway track midway, requiring them to stop, reassess, and then continue, which enabled us to capture their behavior while handling sudden stops and changes in terrain. Navigating open spaces presented another challenge due to the lack of obstacles providing environmental cues, forcing participants to rely heavily on auditory instructions. Walking on curbs involved dealing with intermittent obstacles like parked bicycles, trash cans, and overhanging branches. Ascending and descending staircases further added to the complexity, requiring careful coordination and heightened awareness of their immediate environment. Diverse and realistic scenarios enable BlindWays to capture rich and nuanced motion data, reflecting daily real-world challenges and strategies of blind individuals. Each route is carefully mapped and pre-tested to ensure both feasibility and participant safety. At the start of each route, we provided high-level instructions, including specific objectives and expected challenges. For example, we guided participants by informing them of their current location (e.g., surrounding street names) and the direction they were heading to help them better contextualize the audio navigation aid, which usually guides pedestrians by providing street names and directions. We also briefly explained

potential obstacles they might encounter, such as a train/tram track in the middle of the route or stairs, to prepare them for critical challenges ahead.

**Recording:** We employ the Xsens motion capture system, consisting of 18 Inertial Measurement Units (IMUs) sensors for body joints and a mobility aid, enabling realistic motion capture in various settings. To comprehensively capture the navigation process, we record third-person video of blind pedestrians and egocentric views, as well as motion data. For egocentric views, participants wear a GoPro HERO10 Black on their chest using a comfortable strap, allowing for hands-free and immersive (GoPro Max Lens Mod) recording. The camera is set to face around the participant's feet to meticulously capture cane movements. For third-person views, the accompanying researchers wear a Samsung Galaxy smartphone around the chest and follow the participants without interrupting their natural movements. All data are synchronized, allowing for an in-depth analysis and annotations of navigation strategies and challenges.

To gain further insights into participants' navigation experiences, upon completion of each route, participants are asked to rate their confidence on a scale of 1-7 in (i) their ability to navigate the route and (ii) the guidance that they received from the Google Maps app.

## 3.3 Data Annotation Pipeline

To achieve a nuanced understanding of the navigation behaviors of blind individuals, we employ a meticulous annotation pipeline build in-house that leverages the synchronized third-person view RGB videos along the motion data. To ensure privacy, we mosaic the faces of all people appearing in the videos, both the blind participants and passersby. The annotation process involves 15 human annotators, comprising three motion experts (human biomechanics, sensorimotor, and mobility researchers) and 12 novices, who are provided with detailed instructions, exemplars, and feedback.

Annotators are given 25 videos at a time, and it took approximately two hours to annotate each set of 25 videos. In addition to carefully crafted instructions, novice annotators are also given feedback after the completion of their first set to ensure high quality annotations and help them improve their efficiency in subsequent annotations. Overall, we collect a total of 1,046 videos, highlighting the extensive labor and dedication involved in this annotation process.

To facilitate the annotation process, we build a video annotation interface using Tkinter, a Python-based GUI toolkit. The interface, informed from prior work on video descriptions [45], enables users to freely drag the timeline of the video. Annotators can efficiently review and annotate specific moments in the videos, enhancing the accuracy and detail of their descriptions. For novice annotators, we provided demos of expert annotation samples as references.

**High-Level Descriptions:** For high-level annotations, annotators are requested to focus on describing the overall action of the motion, the purpose behind it, and how the participants were holding their mobility aids (*e.g.*, a cane and a guide dog). Annotators are instructed to provide clear and concise descriptions that convey the intent and broader context of the actions. For example, a high-level description might be: *"A blind man with a cane in his right-hand searches for a street post to press the button. He then orients himself in the direction he wants to cross the street."*

**Low-Level Descriptions:** Low-level annotations require more detailed descriptions of the motion behavior, such as the number of steps taken and the precise use of mobility aids. For instance, a low-level description might be: *"A blind man with a cane searches and locates a street post. He moves forward three steps to orient himself in the direction he wants to cross the street, using his cane in his right hand and positioned in front of him."* The detailed information helps in capturing exact motion dynamics and interactions between the participant and the surrounding dynamic environment. Use of subjective adjectives (*e.g.*, confidently, hesitantly, or meticulously) is encouraged to capture observed behaviors in a more expressive way.

## 3.4 Data Analysis

**Motion Data:** BlindWays captures unique motions characteristic of blind navigation, particularly how individuals use mobility aids to interact with their surroundings in dynamic and complex urban settings. As shown in Fig 2, our motion data include diverse scenarios, from straightforward walking

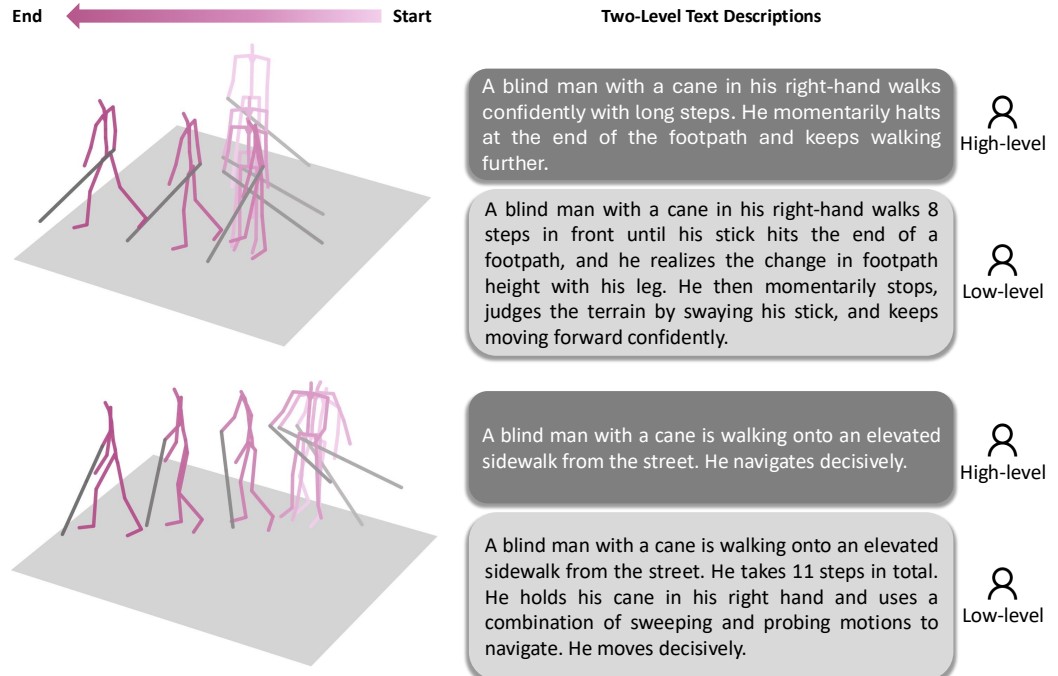

Figure 2: **Qualitative Examples From Our Dataset.** Annotation language captures both high-level information regarding general action, as well as detailed low-level motion characteristics, mobility aid strategies, goals, and environmental context.

behavior with subtle adjustments to avoid obstacles detected by their mobility aids to various turning motions where participants pivot or shift direction using their canes or guide dogs to navigate around corners or obstacles. BlindWays also encompasses scenarios with careful and deliberate movements, such as walking along curbs to avoid falling off the sidewalk or colliding with obstacles, stair navigation with motions like tapping the cane on each step to gauge height and depth and using handrails for support, and street crossing where participants may pause at the curb and perform precise cane movements to locate tactile paving or curb ramps. The data is captured in the Xsens joint representation, comprising a total of 24 joints.

**Text Data:** As shown in Figure 2, high-level motion description annotations provide a summary of the blind person's actions and intent, along with their interactions with mobility aids. Annotations are approximately 26 words long on average, with the longest being 111 words. The standard deviation of about 9.45 indicates a moderate variability in annotation length. In contrast, low-level annotations provide more detailed descriptions of specific actions along with step counts and detailed use of mobility aids. Thus, these annotations are longer, approximately 44 words on average, with the longest being 140 words. The standard deviation of about 17.80 also reflects a higher variability in length, indicative of the varying complexity and level of detail required for different scenarios. Additional details can be found in the supplementary.

## 4  Experiments

In this section, we evaluate human motion generation baselines on BlindWays to discuss model generalizability and the role of text labels in blind motion modeling.

**Metrics and Baselines:** Our evaluation is structured into two parts. First, we discuss fundamental generalization limitations in current text-driven motion generation models, which, despite being built on large motion-language datasets, lack specific categories to capture the unique motions of blind pedestrians. We employ standard metrics used in previous studies: motion-retrieval precision (R Top1) to evaluate the accuracy of matching between texts and motions, Frechet Inception Distance

Table 2: **Text-to-Motion Model Evaluation on BlindWays.** We compare different methods and training datasets for a text-to-motion generation task evaluated on BlindWays. The metrics follow standard text-to-motion evaluation [28]. For the Diversity (DIV) metric, the closer to the results with Real data (first line in the table) the better (indicated with →). Each experiment is repeated 20 times and a statistical interval with 95% confidence is reported.

| Method | Training Set | R Top1 ↑ | FID ↓ | DIV → | MModality ↑ |
|---|---|---|---|---|---|
| Real | - | $0.106^{\pm.0.008}$ | $0.257^{\pm.0.018}$ | $6.232^{\pm.0.258}$ | - |
| HumanML3D [15] | Motion-X [40] | $0.041^{\pm.0.007}$ | $11.203^{\pm.0.109}$ | $5.113^{\pm.0.258}$ | $3.680^{\pm.0.026}$ |
| MotionGPT [28] | Motion-X [40] | $0.046^{\pm.0.006}$ | $15.002^{\pm.0.504}$ | $5.871^{\pm.0.234}$ | $4.646^{\pm.0.171}$ |
| HumanML3D [15] | BlindWays | $\mathbf{0.060}^{\pm.0.012}$ | $\mathbf{3.340}^{\pm.0.257}$ | $5.861^{\pm.0.266}$ | $1.896^{\pm.0.037}$ |
| MotionGPT [28] | BlindWays | $0.054^{\pm.0.008}$ | $5.101^{\pm.0.116}$ | $5.098^{\pm.0.148}$ | $3.993^{\pm.0.134}$ |
| HumanML3D [15] | Motion-X [40] + BlindWays | $0.054^{\pm.0.009}$ | $8.612^{\pm.0.480}$ | $\mathbf{6.260}^{\pm.0.301}$ | $\mathbf{4.921}^{\pm.0.051}$ |
| MotionGPT [28] | Motion-X [40] + BlindWays | $0.036^{\pm.0.003}$ | $10.313^{\pm.0.183}$ | $3.874^{\pm.0.164}$ | $2.759^{\pm.0.100}$ |

(FID) to assess the realism of generated motions, Diversity (DIV) to capture the variance of generated motions, and Multi-Modality (MModality) to examine how generated motions vary within each text description [28]. We adopt HumanML3D [15] and MotionGPT [28] as our state-of-the-art text-to-motion baselines. HumanML3D proposes a framework involving a motion autoencoder, text-to-length, and text-to-motion synthesis, along with a large motion-language dataset for text-driven human motion generation. MotionGPT integrates a generative pre-trained transformer to generate complex motion patterns from text descriptions, leveraging advanced language models.

In the second part of our evaluation, we focus on a motion-driven motion generation task, where we adopt stochastic CVAE-based approaches [33, 34, 73] as our baselines. While these baselines are designed to predict future motion given a history of motion, we further analyze the impact of text descriptions in BlindWays by incorporating a text embedding encoded with LLaMA2 [63]. We employ standard diversity (Average Pairwise Distance, APD) and quality (Average Displacement Error, ADE, and Final Displacement Error, FDE, Normalized Power Spectrum Similarity, NPSS [14], and Normalized Directional Motion Similarity, NDMS [59]) metrics for analysis. We train and test baselines using the joint representation of SMPL (converted from the IMU-based motion capture suit) with an additional joint for the mobility aid of blind pedestrians, ensuring consistency and comparability across experiments and datasets. We split BlindWays into training (85%) and validation/test (15%) sets. For HumanML3D, we use the standard training and validation splits.

**Text-to-Motion:** We provide a comparison of text-to-motion baselines using embedding-based analysis [28] in Table 2. For evaluation, we train a feature embedding model (following HumanML3D [15]). Notably, we observe high FID scores when models are trained exclusively on Motion-X. The lack of blind motion data in Motion-X leads these models to generate diverse yet unrealistic blind motions, increasing the feature space distance between generated and real blind motions. This results in FID scores of 11.203 for HumanML3D and 15.002 for MotionGPT. However, training on BlindWays significantly enhances model performance, with HumanML3D and MotionGPT achieving FIDs of 3.340 and 5.101, respectively. These scores indicate a closer alignment with the real data's FID, reflecting a higher degree of realism. R Precision further underscores the discrepancy in generalizability. We find that the R Top-1 accuracy of models trained on Motion-X is notably lower (0.041 for HumanML3D and 0.046 for MotionGPT) compared to those trained on BlindWays (0.060 for HumanML3D and 0.054 for MotionGPT), indicating that models trained on BlindWays achieve a stronger alignment between text and motion features. In contrast, models trained on Motion-X struggle with this alignment, likely due to the lack of nuanced motion data and corresponding text specific to blind pedestrians. This demonstrates that BlindWays effectively captures the diversity and subtleties of blind pedestrian movements, along with descriptive text, enabling more accurate text-based retrieval and generation.

**Impact of Pre-training:** We examine the impact of pre-training on Motion-X, a large-scale motion-language dataset that covers a broad range of human motions but lacks representations of movements by people with disabilities. As shown in Table 2, pre-training on Motion-X provides the model with a strong understanding of diverse motion features and their alignment with corresponding

Table 3: **Analysis for Specific Keypoints.** We analyze the performance of text-to-motion baselines across different skeleton joint types, including head, arms, and aid. Overall, we find that models pre-trained on Motion-X and then fine-tuned on BlindWays underperform compared to those trained on BlindWays from scratch, particularly in key joints with unique motion distributions in our dataset, such as the arm joints.

| Method | Training Set | All | | Head | | Arms | | Aid | |
|---|---|---|---|---|---|---|---|---|---|
| | | ADE ↓ | FDE ↓ | ADE ↓ | FDE ↓ | ADE ↓ | FDE ↓ | ADE ↓ | FDE ↓ |
| HumanML3D [15] | Motion-X [40] | **3.23** | **3.32** | **0.64** | **0.67** | 0.82 | 0.81 | 0.50 | 0.49 |
| MotionGPT [28] | Motion-X [40] | 3.51 | 3.53 | 0.77 | 0.79 | 0.91 | 0.91 | 0.54 | 0.53 |
| HumanML3D [15] | BlindWays | 3.36 | 3.40 | 0.81 | 0.82 | **0.76** | **0.77** | 0.41 | 0.43 |
| MotionGPT [28] | BlindWays | 3.41 | 3.43 | 0.82 | 0.81 | 0.79 | 0.81 | **0.40** | **0.42** |
| HumanML3D [15] | Motion-X [40] + BlindWays | 3.43 | 3.45 | 0.75 | 0.76 | 0.93 | 0.99 | 0.62 | 0.61 |
| MotionGPT [28] | Motion-X [40] + BlindWays | 3.50 | 3.49 | 0.76 | 0.75 | 1.18 | 1.07 | 0.60 | 0.62 |

Table 4: **Motion Prediction Evaluation on BlindWays.** Given text description and motion history, we predict future 9.5-second 3D poses and compute diversity (APD, higher is better) and quality (ADE, FDE, NPSS, lower is better, and for NDMS, higher values are better) pose metrics.

| Method | APD ↑ | ADE ↓ | FDE ↓ | FID ↓ | DIV → | NPSS ↓ | NDMS ↑ |
|---|---|---|---|---|---|---|---|
| Zero Velocity | - | 0.64 | 0.87 | 12.79 | 3.24 | 1.29 | 0.01 |
| MotionGPT [28] | **23.13** | 3.01 | 4.76 | 0.72 | 4.21 | 0.57 | 0.26 |
| CVAE [34] | 7.68 | 0.47 | **0.56** | 0.45 | 3.89 | **0.11** | 0.23 |
| DLow [73] | 11.65 | 0.46 | 0.59 | 0.41 | 3.91 | 0.12 | 0.27 |
| MDN [33] | 15.14 | **0.45** | **0.56** | **0.40** | **4.31** | 0.14 | **0.28** |

text descriptions. Specifically, HumanML3D shows an average improvement of 6.3 % in Diversity and 4.9% in Multi-Modality when leveraging pre-training on Motion-X, followed by fine-tuning on BlindWays, compared to training on BlindWays alone. This demonstrates the effectiveness of pre-training on a general dataset to enhance motion diversity, even though it lacks categories that capture the unique movements of blind pedestrians.

**Per-Keypoint Evaluation:** We further conduct a per-keypoint evaluation in the text-to-motion task to analyze model performance on blind motion data, with a focus on joints exhibiting unique movements in blind navigation. Specifically, we evaluate the head joint, arm joints (including shoulder, elbow, and wrist, based on participants' dominant hand), and the mobility aid joint, using pose-space accuracy metrics such as ADE and FDE. In blind motion, the arm joints are essential for capturing the use and handling of mobility aids, *e.g.*, for obstacle detection and navigation. The mobility aid keypoint provides insights into the dynamic and coordinated interaction between the user and their aid. As shown in Table 3, we find models trained on BlindWays demonstrate greater robustness, particularly in the arm joint and mobility aid keypoint. Specifically, we observe an average FDE improvement of 8% for arm joints and 16.5% for mobility aid joints in models trained from scratch on BlindWays compared to those trained on Motion-X, highlighting the importance of domain-specific training data for accurately modeling nuanced blind motion behaviors associated with mobility aids. Interestingly, head movements are modeled more accurately by the Motion-X-trained model than the BlindWays-trained model, suggesting that the wider variety of head movements in the Motion-X dataset enhances generalization for these joints. In principle, pre-training on such a dataset could facilitate model generalization for both common behavior joints and unique motion joints. However, in practice, we find this to result in mixed results due to the introduced bias and domain shift. This also highlights a potential direction for future work.

**Motion-Conditioned Prediction:** Finally, we evaluate the capabilities of motion-conditioned models, where both text context and past motion are provided as input to the model. This approach focuses on predicting diverse and plausible future motions given a history of motion. The models are trained to predict the next 9.5 seconds of future motion given 0.5 seconds of past motion. To account for surrounding context-based interactions, we further incorporate text embeddings into stochastic modeling approaches, including CVAE [34], DLow [73], and MDN [33]. For completeness, we incorporate a MotionGPT [28] motion-to-motion baseline and results with a zero velocity model.

We repetitively sample from MotionGPT to obtain APD and pose metrics; however, we note that MotionGPT generally performs poorly in motion-conditioned prediction settings (consistent with the original study of [28]). Table 4 depicts the results for the motion-conditioned models. We find CVAE-based methods to demonstrate better accuracy than MotionGPT, which generally predicts diverse but unrealistic motion patterns. The CVAE baseline achieves an APD diversity of 7.68, while DLow and MDN achieve 11.65 (52% higher) and 15.14 (97% higher) APD, respectively. This finding highlights the benefits of an improved sampling mechanism. Specifically, MDN, which incorporates a transformer-based module in the motion decoding process, significantly enhances both sample diversity and realism (with ADE decreasing from 0.47 to 0.45). We observe consistent improvements in FID, DIV, and NDMS metrics, with MDN achieving the best results. However, for DLow, the increase in diversity is shown to result in an accuracy trade-off, where FDE is increased (from 0.56 to 0.59). While these advancements demonstrate promising strides in modeling diverse and realistic motion, this work represents only an initial step. Future directions include enhancing model generalizability to handle a broader array of rare motion scenarios, such as complex interactions with obstacles or varying terrain, which remain challenging for current models. Additional results and analysis of motion prediction quality within various scenarios can be found in the supplementary material.

## 5 Limitations

Our work addresses a prevalent bias in motion modeling datasets, specifically the focus on sighted and simplified pedestrian motion. Our study underscores the complexity of diverse motion modeling, particularly in cases where pre-training may be non-beneficial or even detrimental to model predictions, such as with blind motion. To tackle this bias, we collected realistically complex data within an important but under-discussed use case. However, our study has several limitations. The sample size of 11 participants, providing a dataset of 1,005 motion samples after filtering pose tracking failure cases, is representative of in-situ accessibility studies. Nonetheless, additional real-world data from a more diverse participant pool could help identify further biases and model issues (e.g., various physical characteristics such as different heights and backgrounds). Another limitation is the expensive ($6,500) motion-capture suit, which may hinder larger-scale studies. While we chose higher-cost, higher-quality tracking technology, lower-cost solutions (e.g., inertial, vision-based) are continuously being developed and can facilitate easier and more scalable capture, leading to more robust and practical motion models across many underrepresented use cases in current human motion benchmarks.

## 6 Conclusion

In this study, we introduce BlindWays, a novel benchmark focused on the unique motion behaviors of blind and low-vision pedestrians navigating dynamic urban outdoor environments. Our dataset includes 3D motion data enriched with high- and low-level text descriptions, derived from corresponding third-person and egocentric RGB videos that capture actions, intentions, and environmental contexts of blind motion in detail—particularly how individuals use mobility aids to interact with their surroundings. Our experiments show that, despite recent advancements, state-of-the-art motion-language models struggle to generalize to blind motion, highlighting the unique challenges presented by this domain. This underscores the importance of a specialized blind motion benchmark to support safe and effective urban planning, such as in autonomous driving. BlindWays provides a contextually rich resource, enabling models to more accurately and diversely represent blind motion, advancing the field of motion-language modeling while enhancing the safety and reliability of real-world, human-interactive systems.

## 7 Acknowledgments

We thank the Rafik B. Hariri Institute for Computing and Computational Science and Engineering (Focused Research Program award #2023-07-001) and the National Science Foundation (IIS-2152077) for supporting this research. Hernisa Kacorri was supported by the National Institute on Disability, Independent Living, and Rehabilitation Research (NIDILRR), ACL, HHS (grant #90REGE0024) and NSF (grant #2229885).

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
