# OpenReview forum: "Text to Blind Motion"
_NeurIPS.cc/2024/Datasets_and_Benchmarks_Track — NeurIPS 2024 Track Datasets and Benchmarks Poster_

### Official Review · Reviewer_c6UN · 2024-07-13
**Review for submission 39**

**Rating:** 6
**Confidence:** 4
**Correctness:** The claims appear to be correct.
**Clarity:** The overall writing is clear.

**Review:**

Pros:

- A motion dataset for individuals with disabilities is meaningful and helpful with positive societal implications.
- The annotation procedure is mostly reasonable.
- Evaluating previous efforts on the proposed dataset helps identify the existing methods' generalization issues.

Cons:

- "Trace and Pace: Controllable Pedestrian Animation via Guided Trajectory Diffusion" (CVPR 2023) could be a better baseline, given its similar focus on pedestrians.
- Why text-to-motion is an important task for BlineWays benchmarking? Justification is needed.
- For text-to-motion, ADE and FDE are inherently limited since they only evaluate the similarity to GT instead of the full-generation ability. Also, why is the diversity metric APD not reported? In addition, the results reported are mixed with marginal differences. Both tune down the fidelity of the accompanying discussions.

**Strengths:**

Please refer to the Review part.

**Additional Feedback:**

- It is mentioned that three experts specialized in different fields were involved in the annotation procedure. However, it is not clear how the experts annotate differently from the novices from the comparison between high-level and low-level descriptions. Is it possible to elaborate more on this?

**Documentation:**

The documentation is sufficient.

**Limitations:**

The limitations are discussed.

**Opportunities For Improvement:**

- As claimed, the motion of blind individuals differs from regular people. Currently, the submissions validate this by the results of generation and prediction tasks indirectly. Direct experimental discussions on this would help us understand this better. Also, a missed chance is to supplement motion from regular people under the same circumstances as a control group to provide more straightforward comparisons.

- The recorded multi-view videos are only adopted for annotation. It is encouraged to explore further the potential applications of visual information.

**Relation To Prior Work:**

The relation to prior works is well-discussed.

**Summary And Contributions:**

The authors introduced the first multimodal blind pedestrian motion benchmark, BlindWays. Wearable sensors are adopted for data collection with multiple subjects and different routes. Textual descriptions are also provided. State-of-the-art models are evaluated, demonstrating limited performance.

---

> ### Author Rebuttal · Authors · 2024-08-17
>
> Thank you for the time and constructive feedback. We have incorporated the suggested analysis, including the additional baseline, direct comparison with sighted motion, and image cues-based ablation, as discussed below. We also provide the requested clarifications regarding task justification and evaluation metrics (a point also raised by Reviewer Z4ZQ, **Q1**).
>
>
> # [Q1] Trace and Pace Baseline
>
> Thank you for the reference, which we now discuss further in our paper (originally cited as [47] in our paper). In contrast to Trace and Pace, we do not assume access to privileged information (e.g., end-point guidance, maps, social information) nor a full physics simulator. Moreover, Trace and Pace does not incorporate text input or report the diversity of generated motion, and only the Trace (i.e., Trajectory diffusion model) component has been previously trained on real-world pedestrian data. We also note that CVAE is the primary baseline in Trace and Pace, which we report in our paper.
>
> To understand the benefits of Trace in our context, we performed an ablation based on the publicly available code. We train the Bird’s Eye View Trajectory diffusion model on BlindWays to compare with our original baselines:
>
> | Model | Train Set | ADE  | FDE  | Velocity | Lon Acc | Lat Acc |
> |-------|-------|-------|-------|-------|-------|-------|
> | Trace | BlindWays  | 0.65 | 1.52 | 0.57 | 0.15 | 0.13 |
> |-|-|-|-|-|-|-|
> | CVAE  | BlindWays  | 0.19 | 0.35 | 0.33 | 0.11 | 0.12 |
> | DLow  | BlindWays  | **0.18** | **0.30** | 0.30 | 0.09 | **0.10** |
> | DLow+  | BlindWays  | 0.19 | 0.33 | **0.27** | **0.05** | **0.10** |
>
>
> | Method | Train Set  | ADE  | FDE  | Velocity | Lon Acc | Lat Acc |
> |-------|-------|-------|-------|--------|-------|-------|
> |Trace | nuScenes | 0.92 | 1.94 | **0.57**| 0.16 | 0.16 |
> |Trace | nuScenes, ETH, UCY | 1.06 | 2.35 | 0.58| 0.18 | 0.19 |
> |Trace | BlindWays | **0.65** | **1.52** | **0.57** | **0.15** | **0.13** |
>
>
> where we implement the metrics from [1] (e.g., Table 5 and Table 6 in [1], similarly not reporting APD). Notably, the model improves accuracy on velocity and acceleration when training on BlindWays compared to a model trained on nuScenes and ETH_UCY, demonstrating the challenging generalization settings. Overall, we show that our studied models outperform the diffusion-based model across metrics in our settings (we also note that related studies also highlight that diffusion-based models provide a trade-off in sample diversity [17]). Unfortunately, incorporating PACE involves significant adaptation, particularly due to the privileged information and full physics simulation required. This is beyond the scope of our work.
>
>
> # [Q2] Why Text-to-Motion?
>
> We have expanded the discussion in Sec. 2 (‘Text-to-Motion’) and Sec. 3 to further justify the benefits of incorporating text. Specifically, as text input provides a controllable interface for generative models, it has gained increased attention [2-7]. In the context of autonomous driving and robotics, text-conditional models can enable a wide range of applications, from data augmentation [4, 7] to counterfactual reasoning and scenario generation [8, 9]. Beyond controllability, our analysis (supplementary Table 6) shows that textual descriptions provide an effective and concise mechanism for incorporating context, and thus improving model performance. We have also incorporated a discussion in Sec. 5 regarding the potential for zero-shot generalization exhibited by multimodal models and its importance in modeling diverse edge cases and long-tail scenarios involving disabled pedestrians.
>
>
>
> # [Q3] Evaluation Metrics
>
> Thank you for the feedback. Please see our response to Reviewer Z4ZQ **[Q1]**.
>
> There are inconsistencies regarding the metrics used in prior work (i.e., pose space-based APD/ADE/FDE in pedestrian-related studies [4, 6-8] vs. feature-based FID/diversity/MultiModality in general motion modeling [9, 5]). We also note that the reference mentioned by the reviewer, Trace and Pace [1], does not report diversity and leverages ADE throughout the analysis. MotionGPT [5] employs different types of metrics for different experiments (e.g., ADE/FDE for motion-to-motion).
>
> In our work, we sought to quantify bias on a **per-joint level** while FID was provided in the supplementary material, Table 3. However, we agree that pose-based metrics can be quite limited on their own and cannot effectively capture the distribution of various motion attributes. We do find FID to improve clarity of trends. APD was originally reported for a subset of the tables, as some focused on biases in accuracy. As suggested by the reviewer, we have improved tables (see Reviewer Z4ZQ, **[Q1]**) to ensure consistent reporting, focusing on FID for overall trends.

---

> > ### Author Rebuttal · Authors · 2024-08-17
> >
> > # [Q4] Additional Ablations
> >
> > **Direct Control Group:** Thank you for the thoughtful feedback. Yes, as essentially all existing benchmarks and models (e.g., HumanML3D, Motion-X, Trace and Pace, others) exclude blind human motion, we have used their predictions as reference. We do not focus on sighted human motion generation. We also note that it is generally understood that blind individuals have different mobility from sighted individuals [10-14] (e.g., in the words of one of our participants, ‘sometimes just finding the crosswalk can be hard’), we see the importance in rigorous comparative analysis.
> >
> > As part of our original study and initial system validation, we did perform data collection with a control group of 10 sighted pedestrians traversing the same routes. However, as our focus is on blind motion generation, these were not annotated with text. Nonetheless, we are happy to incorporate motion-to-motion analysis as a comparison. The discussion has been incorporated in the supplementary material, with results shown in the table below:
> >
> > | Train   | Test    | APD   | ADE   | FDE   | FID  | DIV  | NPSS | NDMS |
> > |-------|-------|--------|-------|-------|-------|-------|-------|-------|
> > | Sighted | Sighted | 11.51 | 0.48  | 0.51  | 0.30  |  3.07 | 0.12 | 0.36 |
> > | Blind   | Sighted | 12.97 | 0.66  | 0.67  | 0.64  |  3.97  | 0.15 | 0.25 |
> > | Sighted | Blind   | 14.30 | 0.70  | 0.78  |  0.92  | 5.13  | 0.15 |0.32 |
> > | Blind   | Blind   | 13.24 | 0.46  | 0.58 | 0.44 | 4.00 | 0.11 | 0.28 |
> >
> >
> > Where we find poor generalization from sighted to blind data (results shown with the DLow+ model).
> >
> > **Role of Visual Cues:** Thank you for this great suggestion. We originally hypothesized that rich text descriptors are sufficient to capture action and scene context. As suggested, we have anonymized the synchronized ego-centric videos and incorporated results using an off-the-shelf visual encoder (SwinTransformer [15]). The extracted features are concatenated with the text embedding. We show complete ablation on input modalities:
> >
> > | Model  | Motion History | Text | Image Context | APD| ADE| FDE| FID| DIV |
> > |-------|-------|-------|-------|-------|-------|-------|-------|-------|
> > | CVAE   | &check; | x | x | 8.80  | 0.47 | 0.60 | 0.48 | 3.92 |
> > | CVAE   | &check; | &check; | x | 7.68  | 0.47 | 0.56 | 0.45 | 3.89 |
> > | CVAE   | &check;| x | &check;| 7.99  | 0.47 | 0.60 | 0.47 | 3.90 |
> > | CVAE   | &check;| &check;| &check;| 8.27  | 0.43 | 0.55 | 0.44 | 4.11 |
> > | DLow   | &check;| x | x | 11.72 | 0.48 | 0.61 | 0.43 | 3.90 |
> > | DLow   | &check;| &check;| x | 11.65 | 0.46 | 0.59 | 0.41 | 3.91 |
> > | DLow   | &check;| x | &check;| 13.10 | 0.47 | 0.60 | 0.43 | 3.98 |
> > | DLow   | &check;| &check;| &check;| 14.62 | **0.42** | 0.54 | 0.40 | 4.53 |
> > | DLow+  | &check; | x  | x | 13.24 | 0.46 | 0.58 | 0.44 | 4.00 |
> > | DLow+  | &check;| &check;| x | 15.14 | 0.45 | 0.56 | 0.43 | 4.18 |
> > | DLow+  | &check; | x | &check;| 16.99 | 0.46 | 0.59 | 0.44 | 4.97 |
> > | DLow+  | &check; | &check;  | &check;  | **17.12** | **0.42** | **0.51** | **0.38** | **4.99** |
> >
> > The ablation demonstrates interesting trends, e.g., we find that our text descriptions provide concise action and surrounding context, outperforming prediction results with visual context alone. However, we find the **text and visual** information to holistically complement, resulting in the best-performing motion prediction model (highest APD, lowest FID). We have integrated this discussion into Sec. 4 (revised Table 4 in the original paper to include results with text+vision+motion input).
> >
> > **Expert vs. Novice:**
> >
> > We have clarified in L. 178 (Sec. 3) that annotation involves an efficient iterative process. First, the experts annotated a small but diverse subset of the videos. After the initial annotations, the experts met with the researchers to standardize descriptions and resolve disagreements among the annotations. These sequences were then provided as examples for training novice annotators. Second, once the novice annotators completed their initial set of annotations, they underwent a corrective feedback round to improve the quality of the annotations. In subsequent annotation sets, if any case of ambiguity occurred throughout subsequent annotations, the samples were sent to an expert for annotation. This process resulted in a standardized annotation process, however, we find that the expert annotators tend to use more expressive and detailed language, while novice users exhibit a more standardized vocabulary:
> >
> > | Annotator | APD |  FID | DIV | Multimodality | NPSS | NDMS |
> > |-------|-------|-------|-------|-------|-------|-------|
> > | Expert | 9.67 | 0.84  | 2.48 |**2.29**|1.00|**0.24**|
> > | Novice |**9.70**|**0.50**|**2.68**| 2.24  |**0.92**|0.23|
> >
> > As shown in the table, existing text encoders can be sensitive to the additional length and vocabulary and details. We have incorporated this discussion in the supplementary material to uncover how predicting intricate motion from such detailed descriptions remains a challenging task (also observed in [7]).
> >
> >
> > We thank the reviewer for the time and suggestions to improve our work. The suggested ablations have been incorporated to strengthen the evaluations and clarity of the paper. Please kindly let us know if there are any follow-up questions or areas needing further clarification

---

> > > ### Author Rebuttal · Authors · 2024-08-17
> > >
> > > ---
> > > \
> > > References
> > >
> > > [1] D. Rempe et al. Trace and pace: Controllable Pedestrian Animation via Guided Trajectory Diffusion. Computer Vision and Pattern Recognition, 2023.
> > >
> > > [2] Shi, X et al., Generating Fine-Grained Human Motions Using Chatgpt-Refined Descriptions. arXiv, 2023.
> > >
> > > [3] Guo, C et al., Generating Diverse and Natural 3D Human Motions From Text.  Computer Vision and Pattern Recognition, 2022
> > >
> > > [4] Zhang, M et al., ReMoDiffuse: Retrieval-Augmented Motion Diffusion Model. ICCV, 2023
> > >
> > > [5] J. Lin et al. Motion-X: A Large-Scale 3D Expressive Whole-Body Human Motion Dataset. Advances in Neural Information Processing Systems, 2024.
> > >
> > > [6] B. Jiang et al. MotionGPT: Human Motion as a Foreign Language. Advances in Neural Information Processing Systems, 2023.
> > >
> > > [7] T. Lee et al. T2LM: Long-Term 3D Human Motion Generation from Multiple Sentences. Computer Vision and Pattern Recognition, 2024.
> > >
> > > [8] S. Tan et al. Language Conditioned Traffic Generation. Conference on Robot Learning, 2024.
> > >
> > > [9] S. Gao et al. VISTA: A Generalizable Driving World Model with High Fidelity and Versatile Controllability. arXiv, 2024.
> > >
> > > [10] M. Williams et al. ‘Just Let the Cane Hit It’ How the Blind and Sighted See Navigation Differently. ACM SIGACCESS Conference on Computers & Accessibility, 2014.
> > >
> > > [11] D. Guth et al. Blind and Sighted Pedestrians’ Judgments of Gaps in Traffic at Roundabouts. Human Factors, 2005.
> > >
> > > [12] D. Ashmead et al. Street Crossing by Sighted and Blind Pedestrians at a Modern Roundabout. Journal of Transportation Engineering, 2005.
> > >
> > > [13] B. Bentzen et al. Wayfinding Problems for Blind Pedestrians at Noncorner Crosswalks: Novel Solution. Transportation Research Record, 2017.
> > >
> > > [14] M. Williams et al. ‘Pray Before You Step Out’ Describing Personal and Situational Blind Navigation Behaviors. ACM SIGACCESS Conference on Computers and Accessibility, 2013.
> > >
> > > [15] Z. Liu et al., Swin Transformer: Hierarchical Vision Transformer Using Shifted Windows. International Conference of Computer Vision, 2021.
> > >
> > > [16] Z. Cao et al. Long-Term Human Motion Prediction With Scene Context. European Conference of Computer Vision, 2020.
> > >
> > > [17] G. Barquero et al. Belfusion: Latent Diffusion for Behavior-driven Human Motion Prediction. International Conference on Computer Vision, 2023.

---

> > > > ### Author Response · Authors · 2024-08-30
> > > >
> > > > Dear Reviewer c6UN,
> > > >
> > > > Thank you once again for taking the time to review our manuscript and for providing valuable feedback. We hope that our rebuttal, which includes requested baselines and reorganized tables, has addressed your questions. If there are any remaining concerns or follow-up questions regarding our analysis, please let us know. Thank you for the suggestions!

---

> > > > > ### Comment · Reviewer_c6UN · 2024-08-30
> > > > >
> > > > > Thanks for the responses. My concerns are addressed. I would update my ratings.

---

### Official Review · Reviewer_Z4ZQ · 2024-07-26
**Review for Text to Blind Motion**

**Rating:** 7
**Confidence:** 4
**Correctness:** The data formation process seems to b…
**Clarity:** The paper is easy to follow

**Review:**

The motion mode of visually impaired persons is highly relevant and under-explored. Adding text descriptions for representation of context is sensible and well motivated. My main concern is the evaluation: the authors provide ample evaluation and ablations of their dataset but only utilize APD, ADE, FDE, which are poor metrics for generative motion (Text2Motion) or (generative) motion forecasting. However, I believe the value of the data to the wider research community outweighs this shortcoming of this work.

The rebuttal addressed my concerns about the evaluation metric.

**Strengths:**

Ground-truth motion representation of the visually impaired is a very important but under-explored motion mode. The dataset does not just provide this different mode of motion but also provides various textural descriptions of the activities of the recorded person, providing a better context.

**Additional Feedback:**

The authors should report the frame rate of their motion dataset

**Documentation:**

The authors promise to setup a GitHub repository for the dataset

**Ethics:**

No concerns

**Limitations:**

Has been discussed

**Opportunities For Improvement:**

For motion prediction APD, ADE and FDE are unconvincing metrics for prediction windows of 9.5s, due to the high degree of stochasticity. Could the authors use the zero velocity model (just repeat the last input pose) as a baseline and report those numbers on ADE and FDE?
I would suggest better evaluation metrics such as NPSS [a] or NDMS [b] for motion quality evaluation. The authors could also report FID score [c], however, I would assume  that this would not work so well, as there is a clear distribution shift between the trained model for FID and this dataset.

Could the authors elaborate how global translation and rotation is handled? This is not clear from the supplementary video.  Are the persons re-normalized at a fixed interval or do they navigate the entire scene in “global coordinates”?

[a] Gopalakrishnan, Anand, et al. "A neural temporal model for human motion prediction." CVPR 2019.
[b] Tanke, Julian, Chintan Zaveri, and Juergen Gall. "Intention-based long-term human motion anticipation." 2021 3DV 2021.
[c] Li, Ruilong, et al. "Ai choreographer: Music conditioned 3d dance generation with aist++." CVPR 2021.

**Relation To Prior Work:**

Has been discussed to some degree

**Summary And Contributions:**

The authors make the point that motion anticipation models are often created in the context of assistance robots or assistance systems. However, the datasets used for training those models do not contain data from persons who would most likely benefit the most from such systems, such as visually impaired persons. The authors compare on two tasks: text-to-motion and motion forecasting, and evaluate on some baseline methods.

---

> ### Author Rebuttal · Authors · 2024-08-17
>
> Thank you for the thoughtful feedback. We are encouraged by the positive remarks on the importance and value of the dataset to the broader research community. As the reviewer rightly states, despite decades of research focused on understanding and modeling humans for assistive and autonomous systems [1-10], there are currently no publicly available 3D motion benchmarks that include visually impaired individuals and are suitable for training and evaluating ML models.
>
> We address the raised concerns regarding additional evaluation metrics and implementation details below.
>
> # [Q1] Additional Evaluation Metrics
>
> Thank you for the references [1, 2, 3], which we have added to Sec. 2. To ensure accurate and comprehensive evaluation, we are happy to revise the analysis and add the requested metrics (FID, NPSS), discussed next.
>
> Although our main contribution lies in a novel multi-modal dataset, as the reviewer notes, there are inconsistencies in the employed metrics by prior work in this domain. For instance, pedestrian-related literature tends to emphasize pose space-based metrics, such as APD/ADE/FDE (e.g., [4, 6-8], where precise prediction is crucial for safety) whereas general human motion generation tends to leverage embedding-based metrics (e.g., FID, diversity, MultiModality [9]). Some studies may choose to mix these; for example, MotionGPT [5] leverages different types of metrics depending on the experiment, such as ADE and FDE (MotionGPT - Table 5) for long-term motion history to motion future prediction.
>
> In our study, we leveraged ADE and FDE to analyze **joint-level performance** and reported FID in the supplementary Table 3, since FID is computed over an embedding and not joints in the pose space. This analysis was **used to highlight motion characteristics within our introduced benchmark** as well as to investigate overall biases in model predictions. However, we agree that pose-based metrics can be quite limited on their own and cannot effectively capture the distribution of various motion attributes.
>
> **Improved Metrics for Motion-to-Motion Results on BlindWays:** As suggested by the reviewer, we have reorganized tables in the main paper to highlight FID and DIV [5], as well as NPSS [1] and NDMS [2] (leveraging publicly available code):
> | Model | APD | ADE | FDE | FID | DIV | NPSS | NDMS |
> |----------|---------|----------|----------|----------|----------|----------|----------|
> | Zero Velocity | - | 0.64 | 0.87 | 12.79 | 3.24 | 1.29 | 0.01 |
> | MotionGPT    |**23.13**| 3.01 | 4.76 | 0.72 | 4.21 | 0.57 | 0.26 |
> | CVAE            | 7.68 | 0.47 |**0.56**| 0.45 | 3.89 |**0.11**| 0.23 |
> | DLow 	    | 11.65 | 0.46 | 0.59 | 0.41 | 3.91 | 0.12 | 0.27 |
> | DLow+	    |15.14 |**0.45**|**0.56**|**0.40**|**4.31**| 0.14 |**0.28**|
>
> We find consistent results, with DLow+ performing well in both FID and diversity. We have also added a discussion on how motion models often provide increased output diversity but at the cost of realism and accuracy, or vice-versa [12]. In general, we find that FID shows clearer trends across evaluation settings (discussed further below), and have added more qualitative results that are aligned with each metric in the supplementary material.
> In the above motion-to-motion results, we repetitively sample from MotionGPT to obtain APD and pose metrics; however, we note that MotionGPT generally performs poorly in motion-conditioned prediction settings (consistent with the original study of [5]). In the table, we have incorporated the suggested zero velocity baseline. We note that for NPSS, lower values are better, and for NDMS, higher values are better.
>
>
>
> **FID Computation:** Yes, there is a distribution shift when training on one dataset and testing on another. When analyzing models trained on BlindWays, we follow MotionGPT [5] and retrain the feature extractor model on BlindWays for evaluation, keeping a similar architecture and definitions as in [5]’s motion-to-motion settings.
>
>
> **Improved Metrics for Text-to-Motion Results on BlindWays:** To better highlight our takeaways, we have improved the tables with the following added metrics:
>
> | Model | Train Set | APD | FID | DIV | MultiModality | NPSS | NDMS |
> |----------|----------|----------|----------|----------|----------|----------|----------|
> | MotionGPT | HumanML3D |17.28|24.35|**9.13**|**7.16**| 0.77 |0.17|
> | MotionGPT | BlindWays|9.79|**0.76**|4.01|3.20|1.00|**0.24**|
> | MotionGPT | HumanML3D + BlindWays|**19.56**|7.13|3.05| 2.69|**0.64**| 0.20 |
>
> Also for **Scenario-Specific Analysis**
>
> | Model | TrainSet | FID - Forward | DIV - Forward | MModality - Forward | FID - Turn | DIV-Turn | MModality - Turn | FID - Sidewalk | DIV - Sidewalk | MModality - Sidewalk | FID - Intersection | DIV - Intersection | MModality - Intersection |
> |-------|-------|-------|--------|-------|-------|-------|-------|-------|-------|--------|-------|-------|-------|
> |MotionGPT|HumanML3D|23.89|**8.65**|**7.74**|22.02|**8.66**|**7.89**|26.09|**4.30**|**7.71**|22.26|**8.60**|**8.04**|
> |MotionGPT|BlindWays|**1.41**|3.74|3.04|**2.14**|3.98|3.46|**1.77**|4.01|3.17|**2.97**|3.53|3.36|
> |MotionGPT|HumanML3D+BlindWays|6.31|3.91|2.74|7.03|2.87|3.09|6.20|3.85|2.87|7.38|3.56|3.11|
>
> In general, we show that model pre-training on other motion datasets, such as HumanML3D, may maintain biases even after fine-tuning. This highlights limited transferability to BlindWays, often resulting in lower FID when training from scratch on BlindWays. On the other hand, models trained on HumanML3D are exposed to other actions (e.g., sitting, dancing) which results in greater overall output diversity.
>
> We thank the reviewer for the suggestions. We have ensured consistent reporting of the suggested metrics in our tables, with pose space-based metrics in the supplementary. Moreover, while each metric differs, we have added a discussion on the need for designing better metrics that holistically evaluate realistic motion models in context [8, 11, 13].

---

> > ### Author Rebuttal · Authors · 2024-08-17
> >
> > # [Q2] Clarification Questions
> >
> > **Global Translation and Rotation:** Our dataset comprises navigating an entire scene in global coordinates. The MoCap sensors can maintain track for 100-200 m ranges, but to ensure high accuracy with minimal drift, our navigation segments were kept under 30m (with re-calibration before each segment). When training the models (including baselines, e.g., MotionGPT), we root normalized coordinates within each frame and predict future pose after normalization by the global coordinate of the root in the initial frame (the origin). We have clarified this in the paper.
> >
> > **Frame Rate:**  We have clarified that the frame rate for the motion data is 60FPS and camera 30FPS.
> >
> > Once again, we thank the reviewer for their time and feedback which helped improve the clarity and impact of our evaluations. We are happy to answer any further questions.
> >
> > ---
> > \
> > References
> >
> >
> > [1] A. Gopalakrishnan et al. A Neural Temporal Model for Human Motion Prediction. Computer Vision and Pattern Recognition, 2019.
> >
> > [2] J. Tanke et al. Intention-Based Long-Term Human Motion Anticipation.  Conference on 3D Vision, 2021.
> >
> > [3] R. Li et al. AI choreographer: Music conditioned 3D Dance Generation with AIST++. Computer Vision and Pattern Recognition, 2021.
> >
> > [4] D. Rempe et al. Trace and Pace: Controllable Pedestrian Animation via Guided Trajectory Diffusion. Computer Vision and Pattern Recognition, 2023.
> >
> > [5] B. Jiang et al. MotionGPT: Human Motion as a Foreign Language. Advances in Neural Information Processing Systems, 2023.
> >
> > [6] Z. Cao et al. Long-Term Human Motion Prediction With Scene Context. European Conference on Computer Vision, 2020.
> >
> > [7] K. Mangalam et al. It Is Not the Journey but the Destination: Endpoint Conditioned Trajectory Prediction. European Conference on Computer Vision, 2020.
> >
> > [8] E. Weng et al. Joint Metrics Matter: A Better Standard for Trajectory Forecasting. International Conference on Computer Vision, 2023.
> >
> > [9] C. Guo et al. Generating Diverse and Natural 3D Human Motions From Text. Computer Vision and Pattern Recognition, 2022.
> >
> > [10] A. Rudenko et al. Human Motion Trajectory Prediction: A Survey. The International Journal of Robotics Research, 2020.
> >
> > [11] D. Dauner et al. NAVSIM: Data-Driven Non-Reactive Autonomous Vehicle Simulation and Benchmarking. arXiv, 2024.
> >
> > [12] G. Barquero et al. Belfusion: Latent Diffusion for Behavior-Driven Human Motion Prediction. International Conference on Computer Vision, 2023.
> >
> > [13] B. Ivanovic and M. Pavone. Injecting Planning-Awareness into Prediction and Detection Evaluation. Intelligent Vehicles Symposium, 2022.

---

### Official Review · Reviewer_S3wn · 2024-07-27
**A good and useful dataset**

**Rating:** 8
**Confidence:** 5
**Correctness:** The paper appears to be correct.
**Clarity:** The paper is in general well written …

**Review:**

Overall, I liked the dataset. The authors have given the necessary description of the dataset along with some benchmarking of its usage. However, what is mostly missing in the paper is a clear description of possible areas where such a dataset will be useful, particularly from the problems that society-centered AI techniques can cater to. Another issue with the paper is that the dataset directory does not contain a clear README about the organization of the dataset.

**Strengths:**

+ A useful dataset of practical significance.
+ Multimodal data with ego-centric video, motion capture and textual description
+ The paper is generally well-written and well-articulated

**Additional Feedback:**

Please check my detailed comments above.

**Documentation:**

The documentation needs further details; specifically, the dataset directory contains only the data files and no specific discussion on the dataset organization.

**Ethics:**

There is no ethical concerns.

**Limitations:**

- The possible research/application areas of this dataset is missing in the paper
- The dataset organization needs to be stated which is missing in the current paper

**Opportunities For Improvement:**

+ Please provide a discussion on how the AI/ML community or pervasive computing community (or anyone else) can benefit from this dataset. More specifically, it would be good to state some research areas where this dataset can be applied.
+ It would be good to provide the details of the dataset organization

**Relation To Prior Work:**

The authors have described the prior works and Table 1 provides a good summary.

**Summary And Contributions:**

This paper discusses a multimodal dataset that captures the 3D movements (wide angle ego camera and motion vectors) of the blind peoples along with textual description of the same. Overall, this is a good dataset that can be applied for several society centric AI problems.

---

> ### Author Rebuttal · Authors · 2024-08-17
>
> Thank you for the valuable feedback and remarks on the significance of our work. We address concerns regarding the dataset's benefits and documentation below.
>
> # [Q1] A Clearer Discussion on the Benefits of the BlindWays Dataset
>
> We see the need to clearly ground our data contribution within societal contexts and real-world applications. As suggested by the reviewer, we have revised Sec. 1 to clarify further that our efforts in modeling the 3D motion of blind pedestrians are motivated by several recent incidents involving autonomous systems and vehicles failing to operate safely around disabled pedestrians, as discussed next.
>
> As a relevant (yet highly concerning) example, we have revised the motivation to highlight Toyota’s suspension of its self-driving fleet at Tokyo’s Paralympic Games, after an accidental collision with a blind athlete in the Olympic Village [1]. This concrete scenario has several important implications for researchers in human- and society-centered machine intelligence, particularly given well-established challenges in model generalization, edge cases, and long-tail prediction (e.g., [2]). Such scenarios may also be challenging for humans; indeed, the person monitoring the aforementioned bus reportedly mentioned that they ‘were aware that a person was there but thought [the person] would [realize that a bus was coming] and stop crossing the street.’ In our work, we asked: would a motion model trained over large amounts of data predominately featuring sighted and typical walking pedestrians behave any differently? Our work takes a foundational step toward addressing such issues, directly impacting and benefiting the urgent application of safety in autonomous systems increasingly deployed in public urban spaces (i.e., through a novel real-world dataset for the ML community).
>
> To further address this important concern, Related Work (Sec. 2) has been improved to strengthen the connection to autonomous driving and robotics. Specifically, we emphasize that gait modeling and motion forecasting are long-studied problems within this context (e.g., [3-7]), yet there is a current gap where blind pedestrians are completely absent from any publicly available real-world benchmarks. Thus, the BlindWays dataset enables ML researchers to explore dataset bias challenges within a novel context.
>
> We foresee other research areas benefiting from our unique multi-modal 3D motion dataset. We have expanded Sec. 5 to discuss potential educational and assistive human-machine interaction applications (these are inherited, to some degree, from general efforts in 3D human motion modeling [8]). These include diverse applications, ranging from enhanced AI interfaces that can support 3D orientation and mobility learning by blind individuals to more inclusive animation generation and simulation that represent the blind community [9, 10]. Our text annotations can directly benefit such applications, i.e., through learning both controllable and accurate predictive models (e.g., Table 6 of the supplementary).
>
> We hope that this revision addresses the raised concern. We agree that this modification improves the clarity of the paper and better highlights our contribution.
>
> # [Q2] Dataset Documentation
>
> Thank you for the suggestion. We are happy to provide additional details on using the dataset. Our original submission linked the data (and Croissant metadata) in the supplementary (L. 18). We have created more detailed resources with documentation and dataset structure explanation, available at
>
> > https://blindways.github.io/
>
> To support ease of use by researchers and practitioners, we are working on adding data visualization and loading scripts, model training tutorials, pre-trained models (with inference demos), and complete evaluation code. Upon paper acceptance and dataset release, we will also maintain a public leaderboard to compare various state-of-the-art models for text-to-motion and motion-to-motion tasks.
> We hope this addresses the concerns regarding organization and dissemination.
>
>
> Once again, we thank the reviewer for the comments and suggestions, which strengthened the impact and clarity of our work. We are happy to incorporate any further suggestions.
>
>
>
> ---
>
> References
>
> [1] The Guardian. Toyota Pauses Paralympics Self-Driving Buses After One Hits Visually Impaired Athlete, 2021.
>
> [2] N. Peri et al. Towards Long-Tailed 3D Detection. Conference on Robot Learning, 2023.
>
> [3] A. Rudenko et al. Human Motion Trajectory Prediction: A Survey. The International Journal of Robotics Research, 2020.
>
> [4] W. Jingbo et al. Learning Human Dynamics in Autonomous Driving Scenarios. Interactional Conference on Computer Vision, 2023.
>
> [5] Z. Yang et al. Recovering and Simulating Pedestrians in the Wild. Conference on Robot Learning, 2021.
>
> [6] S. Ettinger et al. Large Scale Interactive Motion Forecasting for Autonomous Driving: The Waymo Open Motion Dataset. Interactional Conference on Computer Vision, 2021.
>
> [7] D. Rempe et al. Trace and Pace: Controllable Pedestrian Animation via Guided Trajectory Diffusion. Computer Vision and Pattern Recognition, 2023.
>
> [8] J. Lin et al. Motion-X: A Large-Scale 3D Expressive Whole-Body Human Motion Dataset. Advances in Neural Information Processing Systems, 2024.
>
> [9] S. Tan et al. Language Conditioned Traffic Generation. Conference on Robot Learning, 2024.
>
> [10] S. Gao et al. VISTA: A Generalizable Driving World Model with High Fidelity and Versatile Controllability. arXiv, 2024.

---

> > ### Comment · Reviewer_S3wn · 2024-08-27
> >
> > Thanks for the clarifications. The rebuttal clears my doubts and hence, I keep my original score towards accepting the paper.

---

### Author Rebuttal · Authors · 2024-08-17

We thank the reviewers for their time and constructive feedback.

Reviewers note the importance of the introduced text and 3D motion dataset (‘useful dataset of practical significance’, Reviewer S3wn) and the blind motion modeling task (‘very important but under-explored’, Reviewer Z4ZQ). All three reviewers further mention that the paper is well-written, clear, and easy to follow. Reviewer c6UN and Reviewer Z4ZQ also note the value of the dataset for positive societal implications and the wider research community.

We have incorporated the suggestions to improve the paper and addressed concerns regarding the data justification and evaluation metrics below. Please see our reviewer-specific feedback for more information.

---

### Decision · Program_Chairs · 2024-09-26

**Decision:**

Accept (Poster)

**Comment:**

This paper introduces a dataset, which includes 3D motions and their textual descriptions. The dataset can be used for society-centric AI research, which is greatly understudied. Overall, the reviewers are positive about the paper. The area chair recommend accepting this paper.